# Contextualizing Adolescent Female Physical Activity Behavior: A Descriptive Study

**DOI:** 10.3390/ijerph20043125

**Published:** 2023-02-10

**Authors:** Peter Stoepker, Duke Biber, Ashlee Davis, Gregory J. Welk, Adria Meyer

**Affiliations:** 1Department of Kinesiology, Kansas State University, Manhattan, KS 66506, USA; 2Department of Health Promotion & Physical Education, Kennesaw State University, Kennesaw, GA 30144, USA; 3Department of Sport Management, Wellness, & Physical Education, University of West Georgia, Carrollton, GA 75006, USA; 4Department of Kinesiology, Iowa State University, Ames, IA 50011, USA; 5HealthMPowers, Norcross, GA 30071, USA

**Keywords:** population PA, female, health, activity, adolescent, descriptive

## Abstract

Physical activity (PA) behavior tends to decline as youth get older, especially in female adolescents. The purpose of this study was to develop an understanding of female adolescent moderate-to-vigorous physical activity (MVPA) behavior. Baseline MVPA data was collected during year one of a female-specific PA related program. The Youth Activity Profile was administered to contextualize current middle school female PA levels. Data were collected on over 600 6th–8th grade youths with even distributions by grade. No significant differences between grade, race/ethnicity, and MVPA minutes were found. The average estimated value for daily MVPA across all grades was 43.93 (+/−12.97) min, which is considerably lower than the public health recommendation of 60 min per days. Similar amounts were observed for weekend days 45.03 (+/−19.98) and weekdays 45.50 (+/−13.14); however, allocations were smaller during school (9.45 +/− 5.13 min) than at home (34.04 +/− 11.15). The findings from this study highlight the need for further investigation in developing sustainable and innovative PA interventions that target adolescent females.

## 1. Introduction

Current physical activity (PA) guidelines state that children and adolescents need to be active for 60-min every day [1]. Physical inactivity can contribute to the development of several chronic diseases including cardiovascular disease, stroke, type 2 diabetes, certain types of cancer, and obesity [2]. Less than 25% of children and adolescents aged six to seventeen engage in the recommended 60 min of PA per day [3]. Specifically, adolescent females exhibit a decrease in PA as they progress into middle school, with nearly 84% of middle school females engaging in insufficient PA [4]. Regardless of race or ethnicity, research has reported that females are less active than males and that middle school females are less active than high school females [5]. In addition to gender and grade-level differences, racial and ethnic differences in PA have been reported for adolescent females. For example, the decline in adolescent female PA is greater for Black girls than White girls [6]. Furthermore, low levels of PA are an area of concern, as the rate of cardiovascular diseases is rising among adolescent females [7]. If these current trends continue and PA opportunities are not readily available for adolescent females, PA behavior could continue to decline as they enter adulthood [8]. With the historical data reporting declines in PA behavior for adolescent females, it is necessary to continually update reports to accurately reflect PA behavior of each new generation of students. Providing updated descriptive data could lead to an increased awareness of the need to provide PA opportunities for adolescent females.

PA promotion strategies suggest the integration of PA across a variety of settings, including before, during, and after school as well as at home [9]. Out-of-school-time programming has significantly increased over the past few decades and may be an effective setting for evaluating the PA levels of adolescent females [10]. One program that has been developed to inclusively improve adolescent female PA is the Girls Empowering Movement (GEM) initiative [11]. Developed by middle school girls for middle school girls, the GEM initiative aimed to provide out-of-school-time PA led by adult mentors in a socially–emotionally safe environment. The GEM initiative provided moderate-to-vigorous physical activity (MVPA) programming on one day per week for all middle school girls at participating middle schools, with mentors leading programming, PA goal setting, reflective journaling, and mindfulness practice. Prior to the implementation of the GEM initiative, it was necessary to evaluate the baseline PA levels of middle school girls to understand whether the initiative could truly be impactful. The purpose of this study was to examine the baseline daily MVPA of middle school females who were enrolled to participate in the GEM initiative. The results from this study provide contextual data on middle school girl PA behavior, help inform GEM initiative development, and serve as comparative data for long-term GEM evaluation.

## 2. Methods

### 2.1. Study Design

This present study reports on the year one population MVPA and corresponding contextual baseline data collected prior to the implementation of the GEM initiative. This study used a descriptive design to understand current adolescent female MVPA levels. Upon enrollment into the GEM program, participants completed the Youth Activity Profile (YAP) assessment tool to provide baseline data on their current MVPA levels at home, at school, on weekdays, and on weekends. All study procedures were approved by the University of West Georgia Institutional Review Board (protocol code: 21_0112, 16 July 2021).

### 2.2. Participant Recruitment

Purposeful sampling methods were used to recruit participants that were enrolled to participate in the GEM initiative. With permission from each out-of-school time site (*n* = 109 sites), all students who were enrolled in the GEM initiative were eligible to participate in the year one baseline study. In total, a subsample of 619 females across grades 6th–8th completed the YAP survey in baseline year one. Before data was collected, consent from students’ parents/guardians and assent from participants was collected. See Table 1 for a complete breakdown of the study participants. It should be noted that all participants in this study were enrolled to participate in the GEM initiative.

### 2.3. Quantitative Survey

The study utilized the validated online version of the YAP assessment tool [12] to obtain population-level MVPA estimates of females enrolled to participate in the GEM. The YAP is a unique self-report instrument that was designed specifically to produce context-segmented estimates of MVPA behaviors for children in 4th–12th grade [13,14]. Students in the online version of the YAP are guided to complete demographic information (i.e., gender, grade, ethnicity, and race), followed by five items that capture activity at school (transportation to school, during physical education, at recess, in classrooms, and transportation from school) and five items that capture activity at home (before school, after-school, in the evening, on Saturday, and on Sunday). Five additional items evaluate sedentary behavior at home, but these were not analyzed in the study. The focus of this study and the GEM initiative was PA behavior, rather than sedentary behavior and methods to reduce such behavior. Published prediction algorithms are embedded in the software to produce estimates of MVPA for individual items as well as aggregated estimates for MVPA at school, MVPA at home, MVPA on weekdays, MVPA on weekends, and a weighted average for daily MVPA.

### 2.4. Survey Procedures

Students enrolled in the GEM initiative were guided through the YAP instrument by trained GEM program leaders during the first year of GEM implementation. These program leaders completed in-person training conducted by experts who have used the YAP before on how to implement the YAP to their program site participants. Before the distribution of the YAP, a unique username and password was created for each GEM participant so that they could complete the YAP instrument using the online format. Each student was instructed to complete the survey using their own personal login information, with the survey taking approximately 15 min to complete. Although the YAP was completed electronically, the participants completed the instrument at their out-of-school-time site with GEM program leaders present. If middle school participants did not understand a question, the GEM program leaders provided clarification to participants. Participants were not mandated to complete the YAP and were able to withdraw from the study or skip any items of the tool if they desired. Due to the varying start times of each program site and to avoid issues with variability due to seasonality to ensure proper estimates of MVPA behavior, all participating sites had to complete the YAP instrument during the academic school year. Similar population surveillance studies followed similar protocols and analyzed descriptive patterns at the population level during an academic school year [15,16].

### 2.5. Data Analysis

For this study, the primary outcome variables reported were minutes of daily MVPA, weekend MVPA, and weekday MVPA; however, the weekday MVPA values were further segmented into MVPA at home and school time MVPA to provide additional context. Data from these variables were obtained using equations that were specifically calibrated for the YAP tool. Details are available in the published paper [12] or on the online website (www.youthactivityprofile.org). Participants with missing data had to be removed due to the equation used to calculate the output of the YAP tool.

In total, 619 middle school female students completed the YAP during baseline year 1. Prior to analysis, data were screened for missing values, outliers, and violations of statistical assumptions. Students with missing YAP data (*n* = 15) were removed from the data set, resulting in an overall sample of 604 students. First measures of central tendency (e.g., mean) and dispersion (e.g., standard deviation) were completed to contextualize the data set. In addition, multiple one-way ANOVAs were run to examine any potential group differences in grade level or race/ethnicity.

## 3. Results

### PA Findings

A total of 619 6th through to 8th grade female students throughout the state of Georgia participated in the study. Due to missing/incomplete responses, 604 students were included for analysis. There was equal distribution across grade levels (6th grade: *n* = 200, 7th grade: *n* = 204, 8th grade: *n* = 200). There were no significant differences between groups (i.e., grade level, race/ethnicity) in terms of MVPA, with each respective *p*-value being > 0.05. However, descriptive results showed that average MVPA values were less than the daily recommended MVPA values (*M* = 43.93, *SD* = 12.97, *N* = 604). It was not possible to examine individual variability in responses since the YAP is designed for group estimation. However, it is noteworthy that the weekday contributions from school (*M* = 9.45, *SD =* 5.13, *N =* 604) are considerably lower than the contributions from home activities (*M* = 34.04, *SD =* 11.15, *N =* 604). MVPA minutes during the weekend (*M* = 45.03, *SD =* 19.98, *N* = 604) were similar to weekdays (*M* = 45.50, *SD* = 13.14, *N* = 604). Results for each MVPA variable are summarized in Table 2. Lastly, it should be noted that only 12% of females (*n* = 74) enrolled in the GEM initiative met the daily MVPA recommendations at baseline. Although 12% of females met MVPA recommendations, a disproportionate amount (88%) of middle school females reported not meeting daily recommended levels of MVPA.

## 4. Discussion

The current study examined during school, home, weekday, weekend, and daily MVPA behavior in a diverse sample of middle school females enrolled in an out-of-school time program. While there were no significant differences in MVPA in terms of grade and race, the results indicate the overall daily average minutes of MVPA to be lower than recommended levels for this population. A recent population PA surveillance study also found that females and particularly middle school females (i.e., sixth grade) reported lower levels of PA than males [16,17]. Furthermore, our study indicated that there were no statistically significant PA levels based on age/grade level. However, it should be noted that even though it was not significant, students who were in eighth grade reported the lowest daily PA. This finding is consistent with previous studies that found that female PA levels decrease as they move through adolescence [18]. With the goal of helping middle school females achieve the recommended 60 min of MVPA per day, a multicomponent approach (i.e., providing additional PA opportunities outside of the school day) could be used to increase the minutes of MVPA. Specifically, the implementation of before and after school programs have been found to be a promising solution to help children and adolescents engage in health-enhancing MVPA [19]. Recent evidence has indicated that the integration of a before or after school program can have a positive impact on youth PA levels [20], sustained PA participation [21], and overall daily step count [22]. With the potential promise additional PA programming can have on PA behavior, the integration of the GEM initiative will add to the growing body of literature that examines how impactful these programs can be.

The reported MVPA levels for school days, including during school and after school minutes of MVPA, from this study were lower than the recommended amount of MVPA for middle school females in each context [1]. The Society of Health and Physical Educators (Shape America) recommends that middle school students receive 150 min of PA in school per week [23]. The results from this study indicate that middle school females enrolled in the GEM initiative are well below that recommended amount. This could be because Georgia does not mandate middle school physical education. Advocating for policies that require physical education and PA programming at the school level is one method to increase overall PA in middle school [24]. Given the potential impact that integrating before or after school PA programs can have on increasing MVPA, it is important to provide opportunities for middle school females to be physically active before, during, and after school as well as on weekends [25]. The results of study reveal that 88% of middle school girls are not meeting MVPA guidelines, emphasizing the need for out-of-school time programs such as the GEM. While this study provides year one baseline data, it is necessary to evaluate the impact of the GEM initiative following each semester of implementation.

### 4.1. Limitations

There are various limitations to report for this study. First, the MVPA data was collected through self-report measures. Although the YAP is a valid and reliable tool, future studies could collect middle school female MVPA data through accelerometry or other direct measures. Another limitation was the data was only collected at one time point and from middle school females who had registered for an out-of-school time program. School-wide data collection would allow for the greater translation of results. It would also be helpful to collect YAP data at various timepoints throughout the year to detect any changes in MVPA throughout seasons (i.e., the transition from winter to spring), during holidays (i.e., winter break or as students progress into summer break). Despite these limitations, this study adds valuable information on current population PA levels of adolescent females. The strength of this study lies in its use of a valid and reliable surveillance PA tool (YAP) with a large sample of adolescent females (e.g., middle school) to contextualize current MVPA levels.

### 4.2. Implications

The findings of this study are consistent with current population level PA data for adolescent females and continue to highlight the ongoing problem of insufficient PA among adolescents [26]. Even though this study is descriptive in nature, it highlights the continued need to examine PA behavior among adolescent females. Furthermore, the results of this study indicate the need to explore ways to develop innovative strategies that address low levels of PA that target this specific population.

## 5. Conclusions

The purpose of this study was to assess MVPA behavior in middle school females enrolled in an out-of-school time PA program. The results demonstrated that across all the variables measured most students of this specific population of middle school females are not meeting the daily recommendations for MVPA. Due to this, research should continue to explore innovative ways to address PA behavior among adolescent females. Future studies will evaluate the effectiveness of the GEM initiative on middle school girl MVPA levels.

## Figures and Tables

**Table 1 ijerph-20-03125-t001:** Demographic information of participating students.

Grade Level	Total
Sixth	200
Seventh	204
Eighth	200
Race	
Black or African American Caucasian Latina American Indian	52% 34% 11% 1%

**Table 2 ijerph-20-03125-t002:** Average physical activity minutes.

Grade	School PA	Home PA	Weekday PA	Weekend PA	Daily PA
Sixth	9.99 (5.20)	33.59 (11.19)	43.58 (13.29)	43.58 (19.60)	43.58 (12.95)
Seventh	9.61 (5.18)	34.41 (11.49)	44.03 (13.32)	48.04 (19.61)	45.17 (12.76)
Eighth	8.74 (4.95)	34.12 (10.79)	42.87 (12.85)	43.39 (20.46)	43.02 (13.17)
Total	9.45 (5.13)	34.04 (11.15)	45.50 (13.14)	45.03 (19.98)	43.93 (12.97)

Note: Values presented as means (standard deviations).

## Data Availability

Data from the present study can be obtained through emailing the corresponding author: stoepker@ksu.edu.

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
