# Peer review of "Contextualizing Adolescent Female Physical Activity Behavior: A Descriptive Study"

_ijerph, 2023, doi:10.3390/ijerph20043125_

Round 1
Reviewer 1 Report
The article is methodologically correct. The objective is clear and is achieved with research. In my opinion, the results are expected, something that we already know and that continues to be a problem in society. It would be nice to do studies where they ended up talking about realistic proposals to increase the participation of women in physical activity and sport.
The main research question is clear. The theme is not very original because it talks about something that is known. Yes, it is true that it is necessary to corroborate this thought and thus help this practical application.
Regarding the improvements that I propose, it is to increase the sample with other places for example and make a comparison (but this for future publications). What if, is to try to put some more variable so that the results are expanded. Perhaps the section is poorer. It's fine, but if we have to say something, we could correlate it with something else.
The results respond to what is being asked. Not much more can be said, because it is the stated objective. As the conclusion is scarce, it would be good if you make a future intervention or propose what could be done to improve this aspect.
References are correct and current. As for the tables and figures, they comply with the regulations and an index of abbreviations may be missing so that it is clear what each thing is.
Author Response
Thank you very much for taking the time to review our research brief. We appreciate your feedback and insight. Future intervention work will develop based off of these baseline population health results. Our main focus was to provide an updated report on current PA behavior among this specific population. We updated our conclusion to acknowledge that there should be continued work to develop innovative interventions to address adolescent female PA behavior.
Thanks again for your comments and feedback.
Reviewer 2 Report
Thank you for submitting this study. Physical activity participation is an important topic to study. Within your study you analyzed a large group of middle school females in an after-school program within a previous data set. The main goal was to measure students estimated physical activity participation at home, school, on the weekdays, and on the weekends. You looked for differences based on grade level, race/ethnicity, MVPA (weekends/weekdays) and how average scores compared to the recommended amount of daily physical activity. No significant differences were found although daily physical activity was less than the 60 minutes recommended.
Overall, some things are missing or not explained which is why I will not be recommending this article for publication without some revisions. First, the authors did not explain why this study was needed or why they examined the factors that they did. A lack of a literature review also hurts the justification for this study. Some questions I have for the authors would be:
1. Why examine differences in grade level or race? What has previous research found about this that justifies more research on the topic? Or has there been no research on the topic?
2. Are there differences in adolescents' physical activity on weekdays and weekends found in previous studies? Why measure this? I wonder if being in the GEM played a factor in the results being similar?
3. How does this information presented add to the literature on physical activity? It seems like most studies have found that adolescents are below the recommended number of minutes for daily physical activity.
4. What was done during the GEM out-of-school activity? Some information here would be helpful.
5. Why bring up a CSPAP in paragraph two when a literature review is needed much more?
Furthermore, you did not include statistics on race but mentioned them in the results. Could at least be a supplemental file.
Other suggestions/comments:
- Good opening paragraph
- Page (p) 1 Line (L) 37, small literature review needed to help justify the study
- P1L41, Put "physical education" before (PE)
- P3L103, place a period after "estimation."
- P3L118, recommend saying "lower levels of PA than males"
- P4L132, change "have in" to "have on"
- Good limitation paragraph
- P4L146, awkward sentence, I would change ending to "approach to PA on overall PA promotion."
- Overall writing is well done.
Author Response
Thank you very much for taking the time to review our manuscript. We really appreciate the quality review. Attached is a table of how we addressed your specific comments.
Thanks again.

Reviewer 3 Report
The study represents a valid report for future scientific projects.
However, the authors could improve some aspects.
1) improve the study design in materials and methods (possibly incorporate a flow chart)
2) Indicate the most specific statistical analysis: the authors indicate in the text <<In addition, multiple one-way ANOVAs were run to examine any potential group differences on grade level or race/ethnicity>>. What are the relative p-values?
3) In the materials and methods, the authors indicate: Students in the online version are guided to complete 5 items that capture activity at school and 5 items that captureactivity at home (before school, after-school, in the evening, on Saturday, and on Sunday). Five additional items evaluate sedentary behavior at home, but these were not analyzed in the study.
It would be appropriate to report the population as it responded to all 10 items to improve the results in table 2.
Why not try to rank the sample for each item? It would be advisable to add a radar chart with the relative percentages to table 2.
Discussions and conclusions are appropriate
Author Response

(The authors gave the same response as above.)

Round 2
Reviewer 2 Report
Hello again. I really think you did a great job with the edits this time around. I only have one suggestion and one word I would like changed.
Suggestion: In the discussion, I think mentioning that there were no differences in physical activity (PA) based on age/grade level and explaining how this finding relates to previous findings would be appropriate.
Correction needed: Page 5 Line 198, change "level" to "levels." I would also change "brings" to "adds" but I will leave that up to you all.
Author Response
Thank you for your feedback we appreciate the time you have spent to improve our manuscript. Below is how we addressed your feedback.
Suggestion: In the discussion, I think mentioning that there were no differences in physical activity (PA) based on age/grade level and explaining how this finding relates to previous findings would be appropriate.
Thank you very much. Lines 160-164 have been updated to address this comment. It now reads:
Furthermore, our study indicated that there were no statistically significant PA levels based on age/grade level. However, it should be noted even though it was not significant students who were in eighth grade reported the lowest daily PA. This finding is consistent with previous literature that has found that female PA levels decrease as they move through adolescence [18].
Correction needed: Page 5 Line 198, change "level" to "levels." I would also change "brings" to "adds" but I will leave that up to you all.
Thank you so much we changed our language to address these comments. Please see updated lines 201-202. We changed level to levels and brings to adds.
Thank you again for your outstanding feedback.